# *Dianthus superbus* Improves Glomerular Fibrosis and Renal Dysfunction in Diabetic Nephropathy Model

**DOI:** 10.3390/nu11030553

**Published:** 2019-03-05

**Authors:** Jung Joo Yoon, Ji Hun Park, Hye Jin Kim, Hong-Guang Jin, Hye Yoom Kim, You Mee Ahn, Youn Chul Kim, Ho Sub Lee, Yun Jung Lee, Dae Gill Kang

**Affiliations:** 1Hanbang Cardio-Renal Syndrome Research Center, Wonkwang University, 460, Iksan-daero, Iksan, Jeonbuk 54538, Korea; mora16@naver.com (J.J.Y.); hyeyoomc@naver.com (H.Y.K.); aum2668@naver.com (Y.M.A.); host@wku.ac.kr (H.S.L.); 2College of Oriental Medicine and Professional Graduate School of Oriental Medicine, Wonkwang University, 460, Iksan-daero, Iksan, Jeonbuk 54538, Korea; jihuncjstk@naver.com; 3College of Pharmacy, Wonkwang University, Iksan 54538, Korea; mn1003@naver.com (H.J.K.); hg_jin1979@163.com (H.-G.J.); yckim@wku.ac.kr (Y.C.K.); 4School of Pharmacy and Life Sciences, Jiujiang University, Jiujiang 332005, China

**Keywords:** *Dianthus superbus*, diabetic nephropathy, db/db mice, mesangial cell, fibrosis, inflammation

## Abstract

Glomerular fibrosis is caused by an accumulation of intercellular spaces containing mesangial matrix proteins through either diffused or nodular changes. *Dianthus superbus* has been used in traditional medicine as a diuretic, a contraceptive, and an anti-inflammatory agent. The aim of this study was to investigate the effects of *Dianthus superbus*-EtOAc soluble fraction (DS-EA) on glomerular fibrosis and renal dysfunction, which has been implicated in diabetic nephropathy in human renal mesangial cells and db/db mice. DS-EA was administered to db/db mice at 10 or 50 mg/kg/day for 8 weeks. DS-EA treatment significantly ameliorated blood glucose, insulin, the homeostasis model assessment of insulin resistance (HOMA-IR) index, and HbA1c in diabetic mice. DS-EA decreased albumin excretion, creatinine clearance (Ccr), and plasma creatinine levels. DS-EA also ameliorated the levels of kidney injury molecules-1 (KIM-1) and C-reactive protein. DS-EA reduced the periodic acid-Schiff (PAS) staining intensity and basement membrane thickening in glomeruli of the diabetic nephropathy model. In addition, DS-EA suppressed transforming growth factor-β (TGF-β)/Smad signaling. Collagen type IV, a glomerular fibrosis biomarker, was significantly decreased upon DS-EA administration. DS-EA pretreatment attenuated levels of inflammation factors such as intracellular cell adhesion molecule-1 (ICAM-1) and monocyte chemoattractant protein-1 (MCP-1). DS-EA inhibited the translocation of nuclear factor kappa B (NF-κB) in Angiotensin II (Ang II)-stimulated mesangial cells. These findings suggest that DS-EA has a protective effect against renal inflammation and fibrosis. Therefore, DS-EA may serve as a potential therapeutic agent targeting glomerulonephritis and glomerulosclerosis, which lead to diabetic nephropathy.

## 1. Introduction

Diabetic nephropathy is the major representative complication of type 2 diabetes [1]. The major renal characteristics of diabetes include glomerular hypertrophy, persistent albuminuria, loss of podocytes, thickening of the glomerular basement membrane, and matrix expansion [2]. Albuminuria is one of the first asymptomatic clinical features of microvascular damage in diabetes mellitus. These abnormalities result in the leakage of albumin from glomeruli and a progressive decline in renal function with the development of end-stage renal disease accompanied by glomerulosclerosis and fibrosis, ultimately leading to chronic renal dysfunction [3,4]. However, the same pathogenic mechanism has not yet been fully elucidated. Hyperglycemia is known to play an important role in the development of diabetic nephropathy. Additional factors, such as fibrosis-induced renal inflammation, are thought to be important for the initiation and progression of diabetic nephropathy [5]. The major pathological alterations in diabetic nephropathy include mesangial cell proliferation, extracellular matrix expansion, tubulointerstitial injury, and glomerular sclerosis [6].

Transforming growth factor-β (TGF-β) may promote fibrosis through a variety of intracellular signals, such as protein kinases and cytokines: Hence, TGF-β has also been considered as an important regulator of the development of diabetic nephropathy [7]. The active TGF-β1 associates with TβRII (Type II TGF-β receptor), a constitutively active kinase, which further recruits TβRI (Type I TGF-β receptor) and the phosphorylate downstream receptor-associated Smads (R-Smads) Smad2 and Smad3 [8]. The phosphorylated R-Smads form a complex with Smad4 and translocate into the nucleus to regulate the expression of the target genes [9]. TGF-β1 also induces the expression and synthesis of major collagenous components of the extracellular matrix (ECM), such as connective tissue growth factor (CTGF) and collagen IV. Renal fibrosis has been recognized for its critical pathologic features, such as matrix deposition and chronic kidney diseases, as well as for leading to the development of end-stage renal failure and the progression of renal dysfunction [10,11]. CTGF is a highly critical protein associated with structural and functional changes in diabetic renal disease. Recent studies have identified that CTGF plays an important role in promoting fibroblast proliferation, migration, adhesion, and formation of ECM in connective tissue homeostasis [12]. Glomerular CTGF levels are elevated in diabetes-induced animal models, as well as in the renal tissues of diabetic patients. CTGF could promote the deposition of ECM components such as collagen I, collagen IV, and fibronectin, thereby enhancing the disassembly and hypertrophy of mesangial cells [13].

Hyperglycemia-activated inflammation is a major contributor to the pathogenesis of diabetic nephropathy. The inflammation can induce fibronectin expression and ECM accumulation, consequently accelerating the progress of glomerulosclerosis and tubulointerstitial fibrosis [14,15]. Inflammatory chemokines, including monocyte chemoattractant protein-1 (MCP-1) and intracellular cell adhesion molecule-1 (ICAM-1), and high-sensitivity C-reactive protein (hs-CRP) (a marker of inflammation) have been reported to be associated with the development of diabetic nephropathy [16,17]. The activation of NF-κB signaling (the central inflammation pathway) is a key mechanism involving complex inflammatory cytokine regulation in diabetic nephropathy [18].

The C57BL/KsJ-db/db (db/db) mouse serves as a good model for type II diabetes, since it carries a mutation in the leptin receptor gene. This mouse mimics many of the metabolic disturbances of human type 2 diabetes, including glomerular matrix expansion, an increase in renal collagen and serum creatinine, albuminuria, and a reduction in the glomerular filtration rate (GFR), which are similar to advanced human diabetic nephropathy with renal failure [19,20]. Abnormal proliferation of mesangial cells is frequently observed in many diseases that can lead to end-stage renal failure. A variety of early insults, which may be metabolic (as in diabetic nephropathy) or immunological, can lead to uncontrolled mesangial cell proliferation. This in turn causes an increase in ECM deposition and ultimately leads to glomerulosclerosis, resulting in a reduction in the glomerular filtration rate due to a loss of functioning nephrons [21].

*Dianthus superbus* belongs to the group of plants in the family Caryophyllaceae. Traditionally, this plant has been largely used as a diuretic, a contraceptive, and an anti-inflammatory agent, and contains dianthosaponins, dianthramides, flavonoids, coumarin, triterpenoids, pyran-type glycosides, and cyclic peptides [22,23,24]. A previous study has reported that *Dianthus superbus* extracts showed antioxidant, antimicrobial, anticarcinogenic, and anti-inflammatory properties [25,26]. Additionally, *Dianthus superbus* is also known to stimulate immunosuppressive effects, such as osteoblast proliferation and cytotoxic activity against cancer cells [27,28]. However, the effects of *Dianthus superbus*-EtOAc soluble fraction (DS-EA) on diabetic nephropathy have not been reported. Previously, aminoguanidine has ameliorated the development of albuminuria, mesangial expansion, and tissue fluorescence in streptozocin-induced diabetic rats and the overexpression of prosclerotic growth factors and collagen deposition in experimental diabetic nephropathy [29,30]. As a positive control for this study, aminoguanidine was used. Therefore, the aim of this study was to determine whether DS-EA may improve diabetes-associated renal dysfunction, mainly renal fibrosis and inflammation in the diabetic nephropathy model.

## 2. Materials and Methods 

### 2.1. Preparation of Dianthus Superbus

The *Dianthus superbus* L. (1 kg) was prepared from a whole plant and purchased from Daehak Hanyakguk, Iksan, Korea. Dried *Dianthus superbus* (1 kg) was boiled with 10 L of ethanol at 85 °C for 2 h. The extract was centrifuged at 990× *g* for 20 min at 4 °C and the resulting supernatant was collected and lyophilized to powder form (60.1 g). The ethanol extract was dissolved in water (1.5 L), and ethyl acetate (1.5 L) was added. The experimental sample was added to the separatory funnel to dissolve 14.41 g of ethyl acetate fraction, which was then kept at −70 °C until use in this experiment.

### 2.2. Extraction and Isolation

The EtOAc soluble fraction (152.6 mg) was subjected to column chromatography (CC) over a sephadex LH-20 column (Sigma-Aldrich., St. Louis, Mo, USA)) and eluted with a CHCl_3_/MeOH (8:2) system. Fractions were combined based on their TLC pattern to yield subfractions designated E1–E6. Fraction E2 (46.2 mg) was purified by sephadex LH-20 CC (CHCl_3_/MeOH = 9:1) to yield four subfractions (E21–E24). Subfraction E22 (21.6 mg), containing α-spinasterol-3-*O*-β-D-glucopyranoside, was purified by silica gel CC (CHCl_3_/MeOH, 20:1 → 15:1), and finally by silica gel CC (CHCl_3_/MeOH, 18:1), to yield α-spinasterol-3-*O*-β-D-glucopyranoside (5.0 mg).

### 2.3. High Performance Liquid Chromatography (HPLC) Analysis

All high performance liquid chromatography (HPLC) quantitative analysis was carried out using a YL-9100 series HPLC instrument (YL Instruments Co., Ltd., Gyeonggi-do, Republic of Korea) equipped with a sample injector and a PDA detector (YL Instruments Co., Ltd., Gyeonggi-do, Republic of Korea). A column gemini NX-C18 110Å (4.6 mm × 250 mm, 5 μm; Phenomenex Inc., Torrance, CA, USA) was used for chromatographic separation.

#### 2.3.1. Sample Preparation

The concentration of the DS-EA for HPLC analysis, 0.1 mg/20 µL C5D5N, was prepared for quantitative analysis, and 1.2 mg of α-spinasterol-3-*O*-β-D-glucopyranoside was dissolved in C5D5N to make a standard and then diluted with MeOH to 4, 2, and 1 µg/20 µL to construct calibration curves. These samples were filtrated through a 0.45-µm syringe filter before HPLC analysis.

#### 2.3.2. Chromatography Condition

In the HPLC quantitative analysis, the mobile phase was water and acetonitrile, with a gradient elution method: From 0–20 min, a linear gradient from 90% C to 100% C, and from 20–40 min, it was held at 100% C. The flow rate was 0.7 mL/min, the injection volume was set to 20 µL, and the peaks were detected at 195 nm.

### 2.4. Experimental Animals

Male db/db mice (C57BLKS/+Leprdb) and age-matched nondiabetic db/m mice (C57BLKS/J) were purchased at 12 weeks of age from Clea Japan (Tokyo, Japan). The mice were randomly divided into five experimental groups: db/m mice, db/db mice, db/db mice treated with aminoguanidine (AG, 20 mg/kg/day), db/db mice treated with DS-EA-low dose (DSL, 10 mg/kg/day), and db/db mice treated with DS-EA-high dose (DSH, 50 mg/kg/day). Body weight and water/food intake were measured weekly, and blood glucose levels and glucose tolerance tests were measured every 4 weeks in all animals. At 12, 16, and 20 weeks of age, each experimental mouse was placed alone in metabolic cages for 24 h of urine collection. The urine samples were kept at 4 °C until analysis. At the end of the experimental period, all the mice were sacrificed after 12 h of fasting, blood samples were collected into 1-mg/mL ethylenediaminetetraacetic acid (EDTA)-coated tubes after being pulled out of the eyeball, and then the blood samples were maintained at −20 °C until analysis. All procedures were approved by the Institutional Animal Care and Utilization Committee for Medical Science of Wonkwang University (WKU14-58).

### 2.5. Monitoring of Renal Function

Mice of each group were maintained in separate metabolic cages for 2 days, allowing quantitative urine collections and measurements of water/food consumption. Twenty-four-hour urine samples were collected to measure creatinine, osmolality, and other parameters of renal function. In addition, levels of creatinine in plasma were confirmed by a colorimetric method using a spectrophotometer (Milton Roy, Rochester, NY, USA). Osmolality was examined using an Advanced CRYOMATICTM osmometer (Model 3900, Advanced Instruments Inc., Norwood, MA, USA). Urinary albumin was measured at 16 weeks using 24-h collection samples from mice housed in individual metabolic cages. Urine albumin concentration was analyzed by Albuwell (Exocell Inc., Philadelphia, PA, USA). 

### 2.6. Plasma Biochemical Analysis 

Blood samples were taken by periorbital vein for biochemical analysis. Plasma insulin levels were confirmed based on the ELISA method using a commercial mice insulin ELISA kit (Shibyagi Co., Gunma, Japan). Albumin and HbA1c levels in plasma were enzymatically measured using a commercially available kit (ARKRAY, Inc., Minami-Ku, Kyoto, Japan). HbA1c percentage was detected by a fully automated, high-pressure liquid chromatography Tosoh HLC-723 G8 analyzer (Tosoh corp., Tokyo, Japan). Plasma C-reactive protein (CRP) levels were analyzed based on the ELISA method using a commercial mice CRP kit (Biovendor Co., Brno, Czech Republic).

### 2.7. Measurement of Blood Glucose and Insulin Resistance

The concentration of glucose in blood was measured with whole blood samples obtained from a vein in the tail using a Onetouch^®^ Ultra™ Blood glucose Meter (LifeScan, Inc., Milpitas, CA, USA). For the glucose tolerance tests, glucose levels were determined from the tail vein (0 min) before the injection of glucose (1 g/kg body weight). Additional blood samples could be tested at regular intervals (30, 60, 90, and 120 min) for oral glucose tolerance test measurements. Homeostasis model assessment of insulin resistance (HOMA-IR) levels were calculated by the following formula [31]:
$$\begin{array}{ll} HOMA-IR & =\frac{Fasting\;serum\;insulin\;\left(µU/mL\right)\times fasting\;blood\;glucose\;\left(mmol/L\right)}{22.5}\\ & =\frac{Fasting\;serum\;insulin\;\left(ng/mL\right)\times fasting\;blood\;glucose\;\left(mg/dL\right)}{405} \end{array}$$

### 2.8. Quantitative Histopathology

At the age of 16 weeks, the mice were anesthetized and perfused with ice-cold Ringer solution before being perfused and fixed with 10% (*v/v*) buffered formalin in 50-mM potassium phosphate buffer (pH 7.0) for 48 h at 4 °C. For morphometric analysis, kidney tissue slices with 4 μm of thickness were removed and embedded in paraffin and stained with periodic acid-Schiff (PAS) at 400× magnification.

### 2.9. Immunohistochemical Staining

Slides were immunostained by Invitrogen’s Histostain-SP kits using the Labeled-Strept-avidin-Biotin (LAB-SA) method. Slides were immersed in 3% hydrogen peroxide to block endogenous peroxidase activity. Slides were incubated with primary antibodies of collagen IV (Abcam), TGF-β1, ICAM-1, and Nephrin (Santa Cruz Biotechnology, Santa Cruz, CA, USA). All slides were incubated with biotinylated secondary antibodies and horseradish peroxidase-conjugated streptavidin. The detection was visualized using chromogen and counterstaining with 3-6-ethylcarbazole (AEC) followed with hematoxylin (Zymed, Camarillo, CA, USA).

### 2.10. Protein preparation and Western Blot Analysis

Tissue total protein (40 μg) was separated by 10% sodium dodecyl sulfate-polyacrylamide gel electrophoresis (SDS-PAGE), and nuclear and cytoplasmic extracts were prepared on ice as previously described by the method [32]. Protein samples were electrophoresed and then transferred to a nitrocellulose membrane using a Mini-Protean II instrument (Bio-Rad, Hercules, CA, USA). The membranes were blocked with 5% nonfat milk powder in 0.05% Tween 20-phosphate buffered saline (PBS-T), and primary antibodies (Santa Cruz Biotechnology) were incubated at room temperature for 1 h. The blot was incubated with the appropriate horseradish peroxidase-conjugated secondary antibody for 1 h. Protein expression levels were determined by analyzing the signals captured on the nitrocellulose membrane using a Chemi-Doc image analyzer (Bio-Rad).

### 2.11. Cell Cultures

Primary human renal mesangial cells were purchased from ScienCell Corporation (Carlsbad, CA, USA) and cultured in low-glucose Dulbecco’s modified Eagle’s medium supplemented with 10% fetal bovine serum and 1% antibiotic-antimycotic (Gibco, Grand Island, NY, USA). The dispersed mesangial cells were cultured in 95% air and 5% CO_2_ and in a humidified incubator at 37 °C.

### 2.12. Measurement of Cell Proliferation

The effect on primary human renal mesangial cell proliferation of DS-EA was assessed by [3H]-thymidine incorporation. Quiescent cells were treated with 10 μM of angiotensin II and DS-EA, and then 1 μCi of [3H]-thymidine (methyl-[3H] thymidine, 50 Ci/mmol; Amersham, Oakville, ON, Canada) was added for 24 h. Cells were extracted three times with cold 10% trichloroacetic acid (TCA) for 5 min and solubilized for at least 30 min in 0.3 N of NaOH. After harvesting, [3H]-thymidine activity levels were measured using a liquid scintillation counter (Beckman LS 7500, Fullerton, CA, USA). Each experiment was performed in triplicate or quadruplicate.

### 2.13. RNA Isolation and Real-Time PCR

A kit from Qiagen (RNeasy™ Plus mini kit, Qiagen, Inc., Hilden, Germany) was used for RNA isolation from cell cultures, and RNA quality was confirmed by measuring the ratio 260/280 nm using a UV-spectrophotometer. Real-time quantitative RT-PCR reactions and analysis were performed in a 96-well plate by an Opticon MJ Research instrument (Bio-rad) and an optimized standard SYBR Green 2-step qRT-PCR kit protocol (DyNAmo™, Finnzymes, Finland). The primers used were as follows: ICAM-1, sense: 5′-GCT GCT ACC ACA CTG ATG ACG ACA A-3, antisense: 5′-CAG TGA CCA TCT ACA GCT TTC CGG-3′; MCP-1, sense: 5′-GAT CTC AGT GCA GAG GCT CG-3′, antisense: 5′-TGC TTG TCC AGG TGG TCC AT-3′; type IV collagen, sense: 5′-GGT GTT GC A GGA GTG CCA G-3′, antisense: 5′-GCA AGT CGA AAT AAA ACT CAC CAG-3′; CTGF, sense: 5′-GCA AAT AGC CTG TCA ATC TC-3′, antisense: 5′-TCC ATA AAA ATC TGG CTT GT-3′; TGF-β1, sense: 5′-CAA CAA TTC CTG GCG TTA CCT TGG-3′, antisense: 5′-GAA AGC CCT GTA TTC CGT CTC CTT-3′; CTGF, sense: 5′-GCA AAT AGC CTG TCA ATC TC-3′, antisense: 5′-TCC ATA AAA ATC TGG CTT GT-3′; Smad-2, sense: 5′-GTT CAA TCC AGC AAG GAG TAC-3′, antisense: 5′-CTC ATG CGG TGC ACA TTC-3′; GAPDH (a housekeeping gene), sense: 5′-CGA GAA TGG GAA GCT TGT CAT C-3′, antisense: 5′-CGG CCT CAC CCC ATT TG-3′. The real-time PCR data were obtained by using the software provided by the manufacturer.

### 2.14. Reactive Oxygen Species (ROS) Measurement

The fluorescent probe CM-H2DCFDA was used to determine the intracellular generation of ROS by Ang II. Briefly, confluent mesangial cells were pretreated with DS-EA with or without Ang II. The cell was incubated with 10 μM of CMH2DCFDA for 30 min. Fluorescence intensity (relative fluorescence units) was measured at excitation and emission wavelengths of 485 nm and 530 nm, respectively, using a spectrofluorometer (F-2500; Hitachi, Tokyo, Japan). 

### 2.15. Statistical Analysis

Values were expressed as mean ± standard errors (S.E.), and the data were assessed by Sigma 10.0 software (Systat Software, Inc., San Jose, CA, USA). An unpaired Student’s *t*-test was used for comparison between the two groups. For multiple comparisons, data were analyzed by one-way ANOVA with post hoc multiple comparisons. The independent experiments were carried out at least 3 times with similar results. *p* < 0.05 was considered to be statistically significant.

## 3. Results

### 3.1. Calibration Curve and Quantitative Analysis of Standards

Each preparation of DS-EA and the corresponding standards were analyzed using the same HPLC analysis method as described above to calculate the area value for the identified peaks at a retention time of 25.2 min. The calibration curves for the standard were established using the equation *y* = 281.19*x* + 84.721, with correlation coefficients *R*^2^ = 0.9999 (Table 1).

The DS-EA preparation was analyzed using the same HPLC analysis method as above to get the area value of the identified peaks at 25.2 min, as shown in Table 2. In agreement with the data in Table 2 and using the calibration curves equation *y* = 281.19*x* + 84.721, we calculated the amount of α-spinasterol-3-O-β-D-glucopyranoside [33] (Figure 1) in DS-EA. The amount of α-spinasterol-3-O-β-D-glucopyranoside was estimated to be 1.86 μg for 100 μg of DS-EA. In conclusion, there was approximately 1.86% of α-spinasterol-3-O-β-D-glucopyranoside in DS-EA (Table 3).

### 3.2. Effect of DS-EA on Fluid Metabolism in db/db Mice

As shown in Figure 2, treatment of db/db mice with DS-EA demonstrated a significant decrease in the level of food intake, water intake, and urine volume, similarly to levels observed in AG-treated mice, which were taken as a positive control. However, the body weight (BW) and kidney weight % of BW were significantly higher in the db/db groups compared to the control group during 8 weeks of treatment, but there were no significant differences between the db/db mice groups (Appendix A). After 8 weeks of DS-EA treatment, lean mice exhibited a significant reduction in urinary Na^+^, K^+^, and Cl^−^ excretion compared to untreated db/db mice (Appendix A). In contrast, urinary osmolality significantly increased in db/db mice, and was decreased by DS-EA treatment (Figure 2).

### 3.3. Effects of DS-EA on Renal Function and Glomerular Morphology

To determine whether DS-EA treatment can ameliorate renal function-related parameters, we initially analyzed creatinine, urinary albumin, and urea levels in db/db mice. Urinary albumin excretion has been demonstrated to be a good clinical predictor of renal lesions in diabetic nephropathy. As previously reported, the urinary albumin excretion of db/db mice was higher than that of the control, and this further increased with age. This reduction of urinary albumin was maintained throughout the duration of the study. After 8 weeks, the albumin-to-creatinine ratio (ACR) in the db/db group was markedly higher than the control group. The DS-EA 50 mg/kg/day treatment group significantly reduced the ACR level of the diabetic group (*p* < 0.05), similar to the level observed in the AG-treated mice (*p* < 0.05, Figure 3A). The creatinine clearance (Ccr) levels in the db/db mice were significantly reduced in the DS-EA administration group compared to those in the untreated db/db mice (Figure 3B). The urine urea levels in the DS-EA-treated group decreased significantly compared to those in the untreated db/db group (*p* < 0.01, Figure 3C). Additionally, KIM-1, an early biomarker of acute kidney injury, showed a significant decrease after DS-EA treatment in db/db mice (*p* < 0.05, Figure 3D). 

### 3.4. DS-EA Ameliorated Glucose Tolerance and Insulin Resistance in db/db Mice

To understand the effects of DS-EA on glucose metabolism and insulin resistance in db/db mice, we measured fasting blood glucose levels, glucose tolerance, and insulin levels. As shown in Figure 4A, db/db mice exhibited significantly higher blood glucose levels than control mice, which increased consistently for 4 weeks. However, the 10- and 50-mg/kg/day DS-EA treatment groups showed significantly decreased blood glucose levels. In addition, Figure 4B shows that glucose tolerance in DS-EA-treated db/db mice was better than in the untreated db/db group at all tested time periods. Similarly, after treatment with DS-EA, the development of hyperglycemia was alleviated, and the blood glucose and HbA1c at the end of treatment in the 10- and 50-mg/kg/day DS-EA treatment groups were markedly lower than in the untreated db/db mice (Figure 4D, *p* < 0.05 or *p* < 0.01). The HOMA-IR index has been shown to correlate fairly well with invasive tests of insulin sensitivity. As shown in Figure 4E,F, compared to the control mice, plasma insulin levels and the HOMA-IR index increased ~3.5-fold in db/db mice. However, the elevation was substantially reduced by DS-EA treatment (*p* < 0.05).

### 3.5. Effect of DS-EA on Renal Injury and Glomerular Morphological Changes

To determine the effect of DS-EA on kidney structure, particularly in glomeruli, kidney cross-sections were analyzed by PAS staining. PAS staining of the kidneys showed glomerular basement membrane thickening and mesangial expansion as well as increased accumulation of ECM in db/db mice. However, AG or DS-EA treatment ameliorated mesangial expansion compared to the untreated db/db mice (Figure 5). To determine whether DS-EA treatment could improve renal dysfunction through the loss of glomerular nephrin expression, we assessed the expression of nephrin using immunochemistry and western blot analysis. As shown in Figure 5, glomerular nephrin expression in the diabetic kidney decreased significantly, and it was partially restored upon DS-EA treatment. Furthermore, nephrin protein levels were also observed to be upregulated by DS-EA in db/db mice.

### 3.6. Effect of DS-EA on Renal Fibrosis and Inflammation in db/db Mice

As shown in Figure 6, immunohistochemistry and western blot analysis demonstrated that TGF-β1 expression largely increased in the diabetic kidney of db/db mice compared to the control group, whereas TGF-β1 levels decreased significantly in DS-EA-treated db/db mice. DS-EA administration also reduced phosphorylation of Smad-2 and Smad-3 in the kidneys of db/db mice. Next, we examined the extent of glomerulosclerosis by determining the expression of collagen IV, a major ECM protein. Collagen IV expression in db/db mice was significantly higher than in the control mice, and DS-EA treatment markedly lowered collagen IV expression in the db/db mice.

We also examined the effect of DS-EA on diabetes-associated renal inflammation in db/db mice. Immunohistochemistry analysis revealed that DS-EA treatment largely inhibited ICAM-1 expression levels compared to those in the untreated db/db group. Similarly, the levels of pro-inflammatory factors ICAM-1 and MCP-1 also decreased in DS-EA-treated db/db mice compared to the untreated db/db mice. CRP, a biomarker of inflammation, has been widely reported to be associated with the development of diabetic nephropathy. We observed that plasma CRP levels in db/db mice were much higher than in the control group (792.67 ± 207.28 vs. 4748.60 ± 916.56, *p* < 0.05). DS-EA and AG treatment reduced plasma CRP levels in db/db mice (4748.60 ± 916.56 vs. 542.93 ± 93.21, *p* < 0.01) (Figure 6C).

### 3.7. Effect of DS-EA on Human Renal Mesangial Cell Proliferation

Mesangial cell growth plays a critical role in glomerulosclerosis. To determine the effect of DS-EA on the growth of cultured human renal mesangial cells under the effect of Angiotensin II (Ang II), we first assessed the effect of DS-EA (0–50 μg/mL) on mesangial cell viability using 3-(4,5-dimethylthiazol-2-yl)-2,5-diphenyl tetrazolium bromide (MTT) cell proliferation assay. Cell viability assays revealed no obvious cytotoxic effects at 20 μg/ml of DS-EA, but at a concentration of 50 μg/mL, DS-EA showed a 64.5% reduction in cell viability. Thus, in this study, all experiments were performed at concentrations of 10 μg/mL or less (Appendix A). Next, to investigate the inhibitory effects of DS-EA on Ang II-induced mesangial cell proliferation, we performed a [3H]-thymidine incorporation assay. We noted that DS-EA (1–10 μg/mL) inhibited Ang II-induced cell proliferation in a dose-dependent manner (Figure 7A). 

We also examined the effect of DS-EA on cell cycle regulatory proteins by western blot analysis. Both CDK2 and CDK4 were expressed at low levels in control cells, showing an increase after 48 h of stimulation with Ang II. DS-EA treatment effectively inhibited CDK2 and CDK4 expression in a dose-dependent manner. Additionally, the CDK inhibitor p21 (also known as p21WAF1/Cip1), a cell cycle progression-related protein, was upregulated upon pretreatment with DS-EA (Figure 7B).

### 3.8. Effect of DS-EA on Renal Mesangial Cell Fibrosis

To investigate the inhibitory effects of DS-EA on Ang II-instigated mesangial matrix expansion, the production of collagen IV and CTGF was examined using western blot and q-PCR analysis. As shown in Figure 8, the Ang II-mediated increase in fibrogenic collagen IV expression was attenuated by DS-EA in a dose-dependent manner. In addition, the mRNA levels of collagen IV and CTGF were markedly reduced by DS-EA in Ang II-exposed cells. We next examined the effect of DS-EA on TGF-β/Smad signaling in Ang II-exposed human renal mesangial cells. The TGF-β/Smad signaling pathway has been described as playing a critical role in causing glomerulosclerosis and renal fibrosis. We observed that Ang II enhanced the expression of TGF-β, phospho-Smad-2, phospho-Smad-3, and Smad-4 48 h after stimulation. We noted that the expression of fibrosis-related proteins in DS-EA was significantly lower than the expression of Ang II (Figure 8A). In addition, Ang II-induced TGF-β and phospho-Smad-2 mRNA levels decreased significantly upon DS-EA treatment (*p* < 0.01, Figure 8B).

### 3.9. Effect of DS-EA on Renal Mesangial Cell Inflammation

To confirm whether DS-EA was involved in mediating the response to inflammation in mesangial cells, we measured the effect of DS-EA on Ang II-stimulated ICAM-1 and MCP-1 expression. ICAM-1 and MCP-1 protein and mRNA expression levels were enhanced significantly in Ang II-exposed mesangial cells. This elevation was markedly attenuated by the addition of ≥5 μg/mL DS-EA (Figure 9A,B). Next, we examined whether DS-EA regulated Ang II-induced reactive oxygen species (ROS) and NF-κB activation in renal mesangial cells. We observed that Ang II caused NF-κB p65 translocation in the nucleus, which was inhibited by pretreatment with DS-EA (Figure 9C). Intracellular ROS levels were much higher in Ang II-induced cells compared to the control cells. On the other hand, Ang II-induced production of ROS was significantly suppressed by pretreating the cells with ≥5 μg/mL DS-EA in a dose-dependent manner (Figure 9D).

## 4. Discussion

Diabetic nephropathy is one of the typical diabetic complications and is characterized by glomerular hypertrophy and the accumulation of ECM, which eventually leads to glomerular fibrosis. Although current treatments may delay the progression of diabetic nephropathy, the effects on mortality are still limited. This research demonstrated for the first time that DS-EA, a traditional medicine known to have relatively low side effects, ameliorated diabetic nephropathy development by inhibiting mesangial expansion, renal fibrosis, and inflammation. 

The inhibitory effects of DS-EA on diabetic nephropathy are mainly attributed to the amelioration of glucose metabolism through the activation of insulin resistance, which is an important therapeutic strategy for renal disease prevention in type 2 diabetes. The reduction of blood glucose can reduce the risk of developing albuminuria in diabetic patients and mortality in patients with diabetic nephropathy. Therefore, control of blood glucose is one of the current practices of treatment of diabetic nephropathy [34]. Our study indicated that blood glucose levels in untreated db/db mice were significantly higher than in the control group, while the blood glucose of db/db mice, which were treated with DS-EA, showed a decline after 4 weeks of treatment. Glucose tolerance was significantly better in DS-EA-treated db/db mice. In addition, the results of this study show that plasma insulin concentration increased significantly in db/db mice compared to the control group, whereas the insulin levels were reduced by DS-EA treatment. The most widely accepted score for type 2 diabetes mellitus is the HOMA-IR index, which represents the product of glucose and insulin concentrations divided by a factor [35]. DS-EA-treated db/db mice showed a significantly decreased HOMA-IR index. The decrease in plasma insulin/HOMA-IR and the improvement in glucose tolerance suggest that DS-EA may improve insulin resistance. Therefore, insulin resistance is associated with diabetic kidney diseases, and DS-EA prevented diabetic damage to the kidney by improving insulin resistance. Studies have found that HbA1c is an excellent marker of metabolic wellness and a scientifically advanced test that measures the average blood glucose level over the entire previous test period and can accurately evaluate long-term blood sugar management [36]. HbA1c levels were markedly lower in the db/db mice at the end of treatment with DS-EA than in the untreated db/db mice. Thus, these results indicate that DS-EA could reduce blood glucose levels and improve insulin resistance and the HbA1c level in db/db mice, suggesting that DS-EA has a potential role in reducing renal injury in diabetic nephropathy. 

In diabetic nephropathy, the accumulation of ECM components in the glomerular mesangium causes early glomerular hypertrophy and later glomerulosclerosis [37]. Mesangial matrix expansion and the thickening of the glomerular basement membrane have been linked to the progression of renal disease in diabetes mellitus and are hallmarks of diabetic nephropathy [38]. The earliest detectable alteration in the pathogenesis of diabetic nephropathy is an expansion of the glomerular mesangium, which occurs due to excessive accumulation of ECM proteins. PAS staining revealed that db/db mice showed changes in renal morphology, including mesangial matrix expansion, thickening of the glomerular basement membrane, and glomerular hypertrophy compared to the control group. These pathological changes were significantly ameliorated by treatment with DS-EA, suggesting an antifibrotic role of DS-EA in diabetes-associated glomerulosclerosis. Nephrin can directly affect insulin signaling via modulation of glucose transporter vesicle trafficking at the plasma membrane [39]. The expression of nephrin was more reduced in the db/db mice group than in the control group and was also partially recovered by DS-EA treatment. Thus, our results indicate that DS-EA treatment could ameliorate renal dysfunction through its protective effects on podocyte injury via an upregulation of glomerular nephrin expression. In our study, there was sufficient evidence from biochemical parameters and morphologic changes to indicate that DS-EA plays a protective role in renal function in diabetic nephropathy. However, the mechanism of the protective effect of DS-EA on the kidneys is very complex and needs further investigation.

TGF-β1 is a multifunctional cytokine that contributes to a variety of biological processes, including cell proliferation, differentiation, apoptosis, autophagy, and production of ECM. Considerable evidence suggests that TGF-β1 stimulates the transcription of the components of ECM, including collagen, fibronectin, and laminin [40]. TGF-β1 signaling has been shown to be upregulated in injured kidneys and has been suggested to be involved in the development of diabetic nephropathy. In recent years, it has been reported that the amount of collagen IV increases with the progression of diabetic nephropathy in patients and db/db mice. Collagen IV accumulation is a crucial factor in mesangial expansion [41]. Our results showed that DS-EA-treated db/db mice showed reduced expression of TGF-β1 and collagen IV, which have been identified to have an important role in ECM generation. Additionally, Smad-2 and Smad-3 expression was also decreased by treatment with DS-EA. The downregulation of TGF-β signaling pathways may alleviate kidney fibrosis, and our findings demonstrate that DS-EA may have a significant protective effect against diabetic renal injury by regulating the TGF-β signaling pathway.

Hyperglycemia plays an important role in the development of diabetic nephropathy. Additional factors, such as inflammation due to fibrosis, have been considered to be important in the initiation and progression of diabetic nephropathy. Increasing evidence shows that renal inflammation and subsequent fibrosis are critical processes leading to end-stage diabetic nephropathy. Inflammatory cytokines, such as IL-1β and TNF-α, and chemokines, including MCP-1 and ICAM-1, have been shown to contribute to the development of diabetic nephropathy. The activation of NF-κB signaling, the classic inflammation pathway, is a key mechanism involving complex inflammatory cytokine regulation in diabetic nephropathy [42,43]. As expected, the kidneys of db/db mice showed increased expression of ICAM-1 and MCP-1 compared to the control group. The induction of these factors was significantly suppressed by treatment with DS-EA. CRP is a marker of inflammation and has been reported to be associated with the development of diabetic nephropathy [44]. Previous studies have also shown that CRP is induced under diabetic conditions, which in turn synergistically promotes high glucose-mediated renal inflammation and fibrosis both in vitro and in mouse models [45]. Thus, CRP levels have been used as a marker of inflammation. The present study showed that serum CRP levels in untreated db/db mice were induced significantly compared to the control group, while DS-EA-treated db/db mice showed an effective reduction in CRP levels, which was in accordance with the results from previous studies. These results indicate that DS-EA may aid in the improvement of diabetic renal inflammation by downregulating inflammatory mediators.

We performed experiments to determine the possible effects of DS-EA on proliferative, inflammatory, and fibrogenic phenotypes in primary human renal mesangial cells that had been induced by Ang II. Our results indicate that Ang II-mediated stimulation induced renal mesangial cell proliferation and significantly enhanced TGF-β/Smad signaling and collagen IV levels. In addition, Ang II stimulation increased the expression of inflammation-related factors, including ICAM-1, MCP-1, and NF-κB, thereby activating ROS production. Similar to an in vivo study, treatment with DS-EA attenuated renal mesangial cell proliferation by reducing the expression of cell cycle-related proteins. DS-EA treatment decreased Ang II-induced TGF-β/Smad protein and mRNA expression. Moreover, the levels of fibrosis biomarkers, collagen IV levels, and CTGF expression were markedly inhibited by DS-EA treatment. Therefore, we conclude that DS-EA improved Ang II-stimulated renal fibrosis by perturbing TGF-β/Smad signaling. Furthermore, DS-EA suppressed Ang II-induced inflammatory factors and ROS production.

## 5. Conclusions

The in vivo experiments revealed that DS-EA treatment improved diabetes-associated metabolic disorders, such as insulin resistance and renal dysfunction, and markedly attenuated renal inflammation and renal fibrosis in type 2 diabetic db/db mice. Similarly, the in vitro study indicated that DS-EA improved directly Ang II-stimulated mesangial cell proliferation, fibrosis, and renal inflammation in human real mesangial cells. In conclusion, these data provide the first evidence that DS-EA may play, directly or indirectly (through improvement of the glycemic profile), an important role in the prevention of renal fibrosis, inflammation, and hence diabetic nephropathy. Therefore, the protective role of the traditional herbal medicine DS-EA against diabetes-associated renal dysfunction may provide new insights into the development of therapeutic drugs for diabetic nephropathy.

## Figures and Tables

**Figure 1 nutrients-11-00553-f001:**
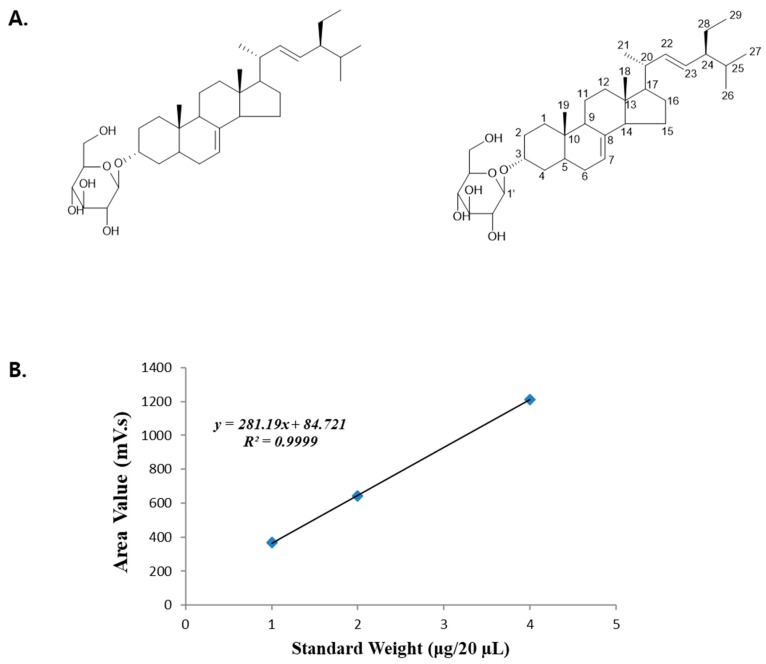
Structure (**A**) and high performance liquid chromatography (HPLC) quantitative analysis (**B**) of α-spinasterol-3-O-β-D-glucopyranoside.

**Figure 2 nutrients-11-00553-f002:**
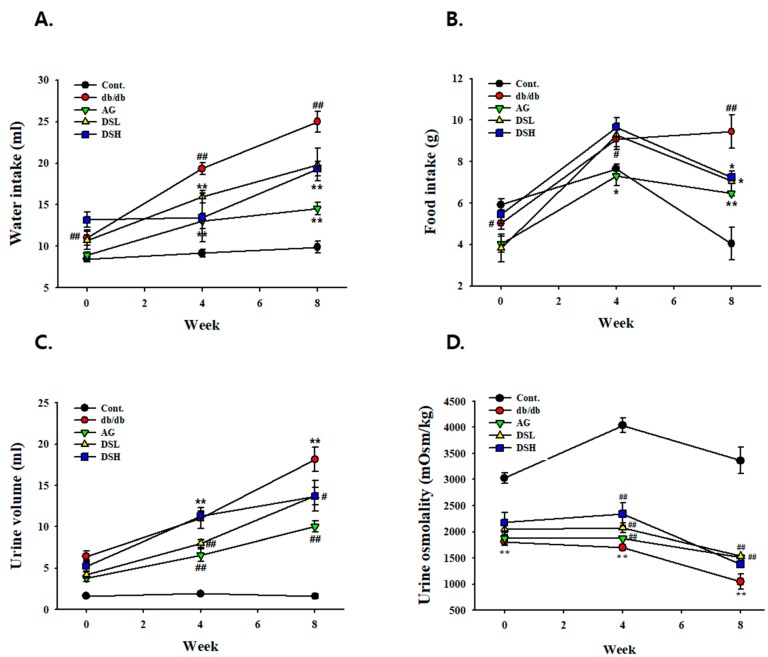
Effects of DS-EA on water intake, food intake (**A**), urine volume (**B**), urine volume (**C**), and urine osmolality (**D**). Values are expressed as mean ± S.E. (*n* = 8). ** *p* < 0.01 vs. Cont.; # *p* < 0.05, ## *p* < 0.01 vs. negative Cont.; Cont.: db/m mice group; db/db: db/db mice group (negative Cont.); AG: db/db mice treated with aminoguanidine group; DSL: db/db mice treated with low-dose DS-EA group; DSH: db/db mice treated with high-dose DS-EA group; S.E., standard errors.

**Figure 3 nutrients-11-00553-f003:**
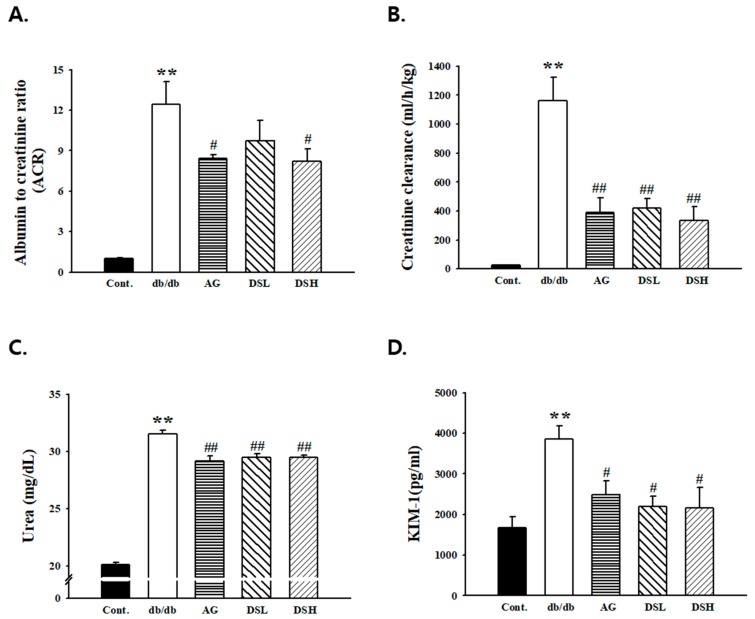
Effect of DS-EA on the albumin-to-creatinine ratio (ACR) (**A**), creatinine clearance (**B**), urine urea (**C**), and KIM-1 (**D**). Values are expressed as mean ± S.E. (*n* = 5). ** *p* < 0.01 vs. Cont.; # *p* < 0.05, ## *p* < 0.01 vs. negative Cont.; Cont.: db/m mice group; db/db: db/db mice group (negative Cont.); AG: db/db mice treated with aminoguanidine group; DSL: db/db mice treated with low-dose DS-EA group; DSH: db/db mice treated with high-dose DS-EA group.

**Figure 4 nutrients-11-00553-f004:**
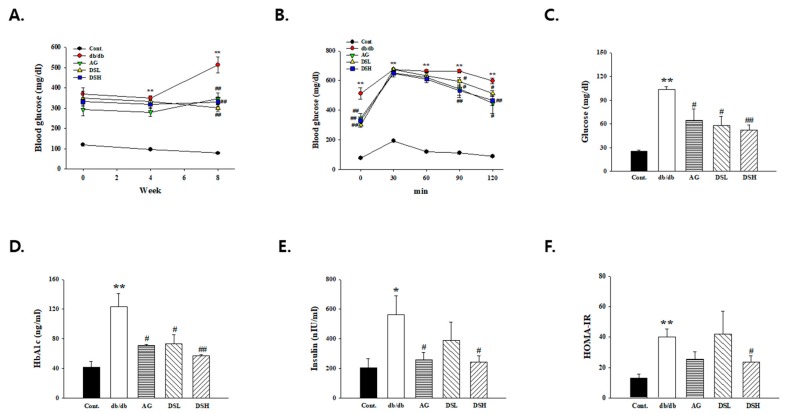
Effect of DS-EA on blood glucose (**A**), oral glucose tolerance test (**B**), glucose (**C**), HbA1c (**D**), insulin (**E**), and HOMA-IR (**F**). Values are expressed as mean ± S.E. (*n* = 5). * *p* < 0.05, ** *p* < 0.01 vs. Cont.; # *p* < 0.05, ## *p* < 0.01 vs. negative Cont.; Cont.: db/m mice group; db/db: db/db mice group (negative Cont.); AG: db/db mice treated with aminoguanidine group; DSL: db/db mice treated with low-dose DS-EA group; DSH: db/db mice treated with high-dose DS-EA group.

**Figure 5 nutrients-11-00553-f005:**
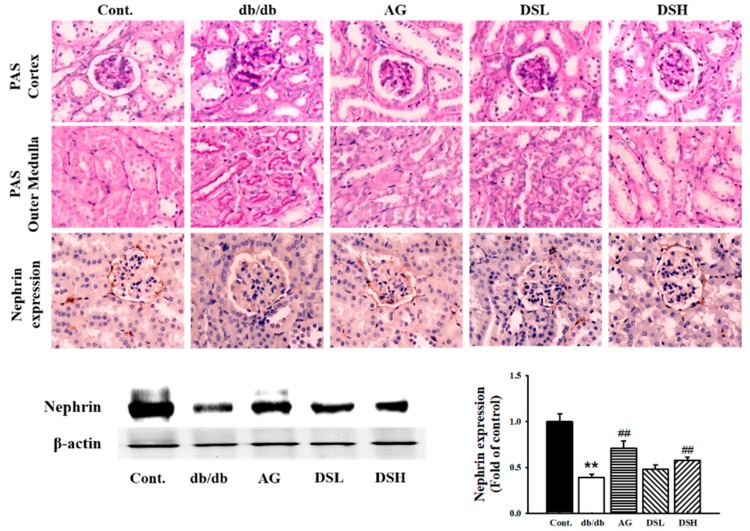
Effect of DS-EA on renal pathological changes and nephrin immunoreactivity in db/db mice. Representative microscopic photographs of a kidney stained with periodic acid-Schiff (PAS). Kidney sections in the cortex (glomerulus) and outer medulla were obtained from the db/m (Cont.) group, db/db (db/db) group, aminoguanidine-treated db/db group (AG), DS-EA low-dose-treated db/db group (DSL), and DS-EA high-dose-treated db/db group (DSH). The expression of nephrin in the kidneys was determined by immunohistochemistry staining or Western blot analysis (*n* ≥ 3, magnification 400×). ** *p* < 0.01 vs. Cont.; ## *p* < 0.01 vs. negative Cont.; Cont.: db/m mice group; db/db: db/db mice group (negative Cont.); AG: db/db mice treated with aminoguanidine group; DSL: db/db mice treated with low-dose DS-EA group; DSH: db/db mice treated with high-dose DS-EA group.

**Figure 6 nutrients-11-00553-f006:**
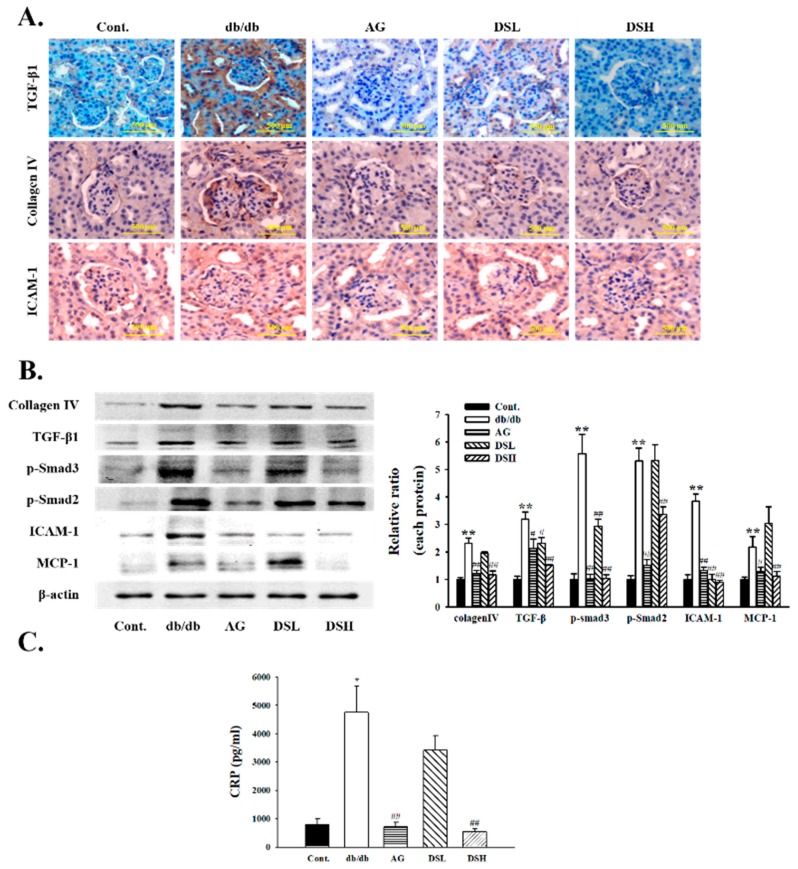
Effect of DS-EA on glomerular fibrosis in the cortex of kidney tissues. (**A**) Immunohistochemistry detected that DS-EA therapy inhibited TGFβ1, collagen IV, and ICAM-1 expression in the diabetic kidney of db/db mice (*n* = 3, magnification 400×). (**B**) Expression of protein in mouse kidneys was determined by western blot analysis (*n* ≥ 3). (**C**) Plasma C-reactive protein (CRP) levels were measured at wavelengths of 450 nm using an ELISA reader. Values are expressed as mean ± S.E. (*n* = 8). * *p* < 0.05, ** *p* < 0.01 vs. Cont.; # *p* < 0.05, ## *p* < 0.01 vs. negative Cont.; Cont.: db/m mice group; db/db: db/db mice group (negative Cont.); AG: db/db mice treated with aminoguanidine group; DSL: db/db mice treated with low-dose DS-EA group; DSH: db/db mice treated with high-dose DS-EA group.

**Figure 7 nutrients-11-00553-f007:**
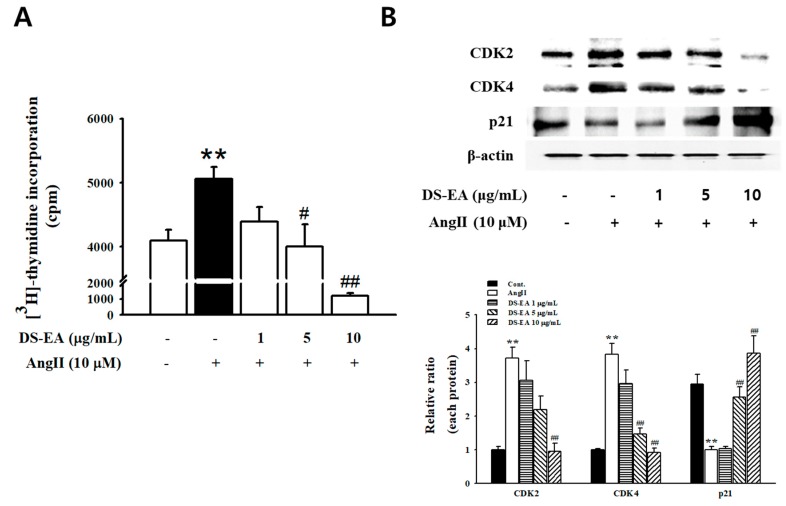
Effect of DS-EA on Angiotensin II (Ang II)-induced human renal mesangial cell proliferation. (**A**) Mesangial cells were seeded into 24-well plates. After confluence, the cells incubated for 48 h with or without Ang II and various concentrations of DS-EA (1–10 μg/mL) and were then pulse-labeled with [3H]-thymidine for 24 h. (**B**) Western blot analysis was performed with antibodies specific for CDK2, CDK4, and p21waf1/cip1. Results are expressed as the mean ± S.E. from five independent experiments. ** *p* < 0.01 vs. control; # *p* < 0.05, ## *p* < 0.01 vs. Ang II alone.

**Figure 8 nutrients-11-00553-f008:**
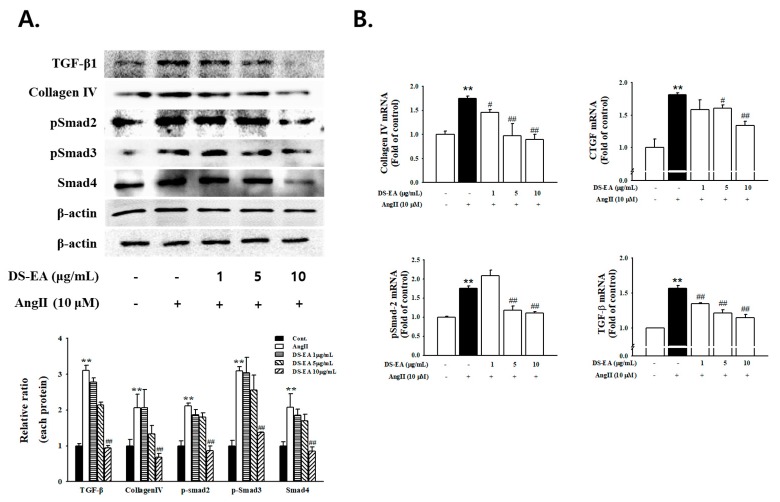
Effect of DS-EA on glomerular fibrosis in human renal mesangial cells. (**A**) Cell lysates were used for western blot analysis with a primary antibody against TGF-β1, collagen IV, pSmad-2, pSmad-3, and Smad-4. The protein bands were detected by western blotting, and β-actin was used as the internal standard in each sample. (**B**) Real-time PCR showing mRNA levels in DS-EA-treated and Ang II-stimulated mesangial cells. Each value represents the mean ± S.E. of five independent experiments. ** *p* < 0.01 vs. control; # *p* < 0.05, ## *p* < 0.01 vs. Ang II alone.

**Figure 9 nutrients-11-00553-f009:**
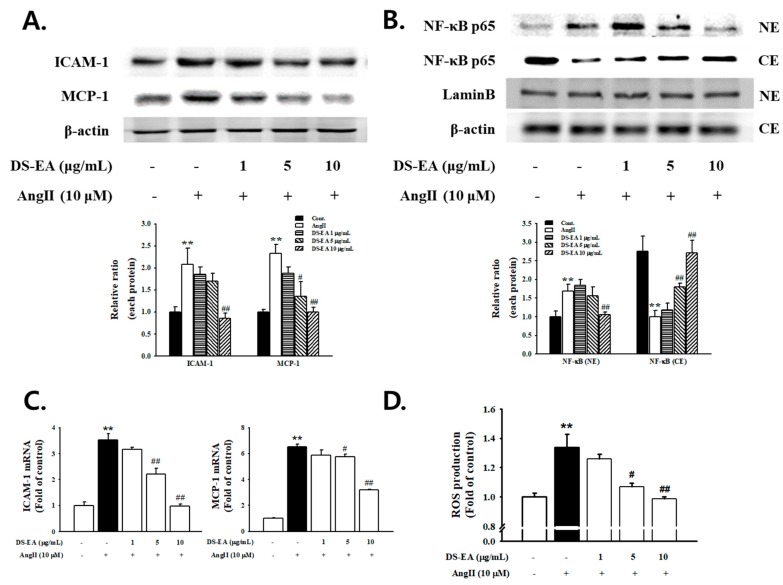
Effect of DS-EA on renal inflammation in human renal mesangial cells. Cells were pretreated with DS-EA for 30 min and then stimulated with Ang II (10 μM) for 48 h. (**A**) Whole protein was used for western blot analysis with a primary antibody against ICAM-1 (85-110 kDa) and MCP-1 (12 kDa). In addition, β-actin was used as an internal control. (**B**) RT-PCR showed mRNA levels of ICAM-1 and MCP-1. The effect of DS-EA on Ang II-induced NF-κB translocation (**C**) and reactive oxygen species (ROS) production (**D**) in mesangial cells. For western blot analysis with a primary antibody against NF-κB p65, nuclear fractions were obtained. Lamin B (67 kDa) was used as an internal control. Each value represents the mean ± S.E. of five independent experiments. ** *p* < 0.01 vs. control; # *p* < 0.05, ## *p* < 0.01 vs. Ang II alone.

**Table 1 nutrients-11-00553-t001:** High performance liquid chromatography (HPLC) quantitative analysis results of α-spinasterol-3-O-β-D-glucopyranoside.

Standard Weight (μg/20 µL)	Area Value (mV·s)
4	1210.772
2	643.183
1	368.516

**Table 2 nutrients-11-00553-t002:** HPLC quantitative analysis results of *Dianthus superbus*-EtOAc soluble fraction (DS-EA).

DS-EA (μg/20 µL)	Area Value (mV·s)	Contents of Standard (μg)	Retention Time (min)
100	607.863	1.86	25.2

**Table 3 nutrients-11-00553-t003:** The contents of α-spinasterol-3-O-β-D-glucopyranoside in *Dianthus superbus*-EtOAc soluble fraction (DS-EA).

Retention Time (min)	Contents in DS-EA (*w/w*)
25.2	1.86%

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
