# Peer review of "Dianthus superbus Improves Glomerular Fibrosis and Renal Dysfunction in Diabetic Nephropathy Model"

_nutrients, 2019, doi:10.3390/nu11030553_

Reviewer 1 Report

The authors of the study investigated the renoprotective effects of the ethanol-ethyl acetate extract of the plant Dianthus superbus in vivo (using a db/db mice model) and an in vitro (Angiotensin II-induced mesangial cell injury). The study concludes that Dianthus superbus improves insulin sensitivity, which corrects hyperglycemia and ameliorates the deleterious effects mediated by hyperglycemia on the kidney. In addition, in vitro studies in mesangial studies suggest that Dianthus superbus also exhibits direct renoprotective effects presumably via its anti-inflammatory (ICAM-1, NF-kappa B signaling), anti-fibrotic (Collagen IV, TGF-beta signaling) and antioxidant effects. 

Spelling and Grammar

Several spelling errors were noted on the manuscript. E.g. Line 201: Qiagen is misspelled as Qiagene; Line 322: Magnification is misspelled as magrification.

Methods

1) It is not unclear whether the extract was prepared from the whole plant, stem, bark or leaves. This needs to be clearly specified in the methods section. 

2) The rationale for the use of an aminoguanidine-treated control (db/m) mice group in the study.

3) Control mice treated with the low-dose and high-dose of the extract (db/m+DSL and db/m+DSH) is missing in the experimental design. These two groups are crucial for comparing the effects of DSL and DSH in db/db mice. 

4) Similarly, the aminoguanidine-treated db/db mice group is also missing. 

5) The formula for HOMA-IR is missing in the section 2.7. Line 162. 

Results

1) Figure 3A. Unlike the plasma creatinine, representation of urine creatinine doesn’t add much value as it can vary quite significantly in an individual or animal. The data should be represented as Urine Albumin-to-Creatinine-ratio (UACR).

2) Figure 3D. The reason for a lesser reduction in the UAE in DSH as compared to DSL needs to be explained in the discussion. 

3) In general, the abbreviations in all the figure legends need to be expanded.

4) Figure 5A. The PAS-stained sections and IHC micrographs need to be shown at higher magnification to reveal differences across the groups. 

5) Figure 5. The western blot for Nephrin expression shown in the manuscript doesn’t indicate any difference between db/db and DSH group. A more representative western blot needs to be shown to indicate the changes observed with DSH.

6) Similar to Figure 5A, the micrographs shown in Figure 6A needs to be magnified to reveal differences across the groups. 

5) Figures 6, 7, 8, and 9. The densitometry graphs for all western blots are required in addition to the blots. 

Major Concern

The lack of three critical groups - Control mice treated with the low-dose and high-dose of the extract (db/m+DSL and db/m+DSH) and aminoguanidine-treated diabetic group - in the in vivo study raises a major concerns because it makes it hard to corroborate the effects of DS extract at low and high doses on control mice. Hence, this reviewer recommends including these groups to assure the scientific validity of the findings from the study. 

Author Response

I am very much thankful to the reviewers for their deep and thorough review. I hope my revision has improved the paper to a level of their satisfaction. I revised the text in the Methods, Results, Discussion, and reference (red colored part).

Reviewer 1

The authors of the study investigated the renoprotective effects of the ethanol-ethyl acetate extract of the plant Dianthus superbus in vivo (using a db/db mice model) and an in vitro (Angiotensin II-induced mesangial cell injury). The study concludes that Dianthus superbus improves insulin sensitivity, which corrects hyperglycemia and ameliorates the deleterious effects mediated by hyperglycemia on the kidney. In addition, in vitro studies in mesangial studies suggest that Dianthus superbus also exhibits direct renoprotective effects presumably via its anti-inflammatory (ICAM-1, NF-kappa B signaling), anti-fibrotic (Collagen IV, TGF-beta signaling) and antioxidant effects. 

Spelling and Grammar

Several spelling errors were noted on the manuscript. E.g. Line 201: Qiagen is misspelled as Qiagene; Line 322: Magnification is misspelled as magrification.

- We corrected it

Methods

1) It is not unclear whether the extract was prepared from the whole plant, stem, bark or leaves. This needs to be clearly specified in the methods section. 

- We added it in method part

 2) The rationale for the use of an aminoguanidine-treated control (db/m) mice group in the study.

-  We corrected and have made the following changes and add the rationale for the use of an aminoguanidine-treated control db/m mice treated with Aminoguanidine (AG, 20 mg/kg/day) → db/db mice treated with Aminoguanidine (AG, 20 mg/kg/day)

 3) Control mice treated with the low-dose and high-dose of the extract (db/m+DSL and db/m+DSH) is missing in the experimental design. These two groups are crucial for comparing the effects of DSL and DSH in db/db mice. 

- This experiment confirmed only the improvement of renal fibrosis in the Dianthus superbus diabetic nephropathy model. In the future, we will conduct further using experiment model in treated with the low-dose and high-dose of the extract (db/m+DSL and db/m+DSH).

 4) Similarly, the aminoguanidine-treated db/db mice group is also missing. 

- We corrected it .

 5) The formula for HOMA-IR is missing in the section 2.7. Line 162. 

- We added it

Results

1) Figure 3A. Unlike the plasma creatinine, representation of urine creatinine doesn’t add much value as it can vary quite significantly in an individual or animal. The data should be represented as Urine Albumin-to-Creatinine-ratio (UACR).

- We added figure of Urine Albumin-to-Creatinine-ratio (UACR).

 2) Figure 3D. The reason for a lesser reduction in the UAE in DSH as compared to DSL needs to be explained in the discussion. 

- The DSL and DSH data Item are wrong in Figure 3D. Therefore, we corrected it by mistake.

 3) In general, the abbreviations in all the figure legends need to be expanded.

- We corrected it in all the figure legends

 4) Figure 5A. The PAS-stained sections and IHC micrographs need to be shown at higher magnification to reveal differences across the groups. 

- We changed the Figure 5A to a higher magnification.

 5) Figure 5. The western blot for Nephrin expression shown in the manuscript doesn’t indicate any difference between db/db and DSH group. A more representative western blot needs to be shown to indicate the changes observed with DSH.

- We corrected it

 6) Similar to Figure 5A, the micrographs shown in Figure 6A needs to be magnified to reveal differences across the groups. 

- We changed the Figure 6A to a higher magnification.

 7) Figures 6, 7, 8, and 9. The densitometry graphs for all western blots are required in addition to the blots. 

- We added the densitometry graphs for all western blots.

Major Concern

The lack of three critical groups - Control mice treated with the low-dose and high-dose of the extract (db/m+DSL and db/m+DSH) and aminoguanidine-treated diabetic group - in the in vivo study raises a major concerns because it makes it hard to corroborate the effects of DS extract at low and high doses on control mice. Hence, this reviewer recommends including these groups to assure the scientific validity of the findings from the study. 

 - We accepted your revision. We have focused therapeutic effect of natural extract or fraction in various renal dysfunction animal models. In previous studies, we had only done a comparative experiment between db / db models (1-3). In addition, other studies were performed by db/db mice for the search the effective herbs for the treatment renal diseases (4-9). Thus, we have a confidence our experimental design to clarify the beneficial effect of Dianthus superbus targeting diabetic nephropathy. Later, the further pharmacological efficiency of the DS extract will be clarified through comparative experiments between the db/m groups. 

1. Lee AS, Lee YJ, Lee SM, Yoon JJ, Kim JS, Kang DG, Lee HS. Portulaca oleracea Ameliorates Diabetic Vascular Inflammation and Endothelial Dysfunction in db/db Mice. Evid Based Complement Alternat Med. 2012;2012:741824.

2. Hwang SM, Kim JS, Lee YJ, Yoon JJ, Lee SM, Kang DG, Lee HS. Anti-Diabetic Atherosclerosis Effect of Prunella vulgaris in db/db Mice with Type 2 Diabetes. Am J Chin Med. 2012;40(5):937-51.

3. Lee AS, Lee YJ, Lee SM, Yoon JJ, Kim JS, Kang DG, Lee HS. An Aqueous Extract of Portulaca oleracea Ameliorates Diabetic Nephropathy Through Suppression of Renal Fibrosis and Inflammation in Diabetic db/db Mice. Am J Chin Med. 2012;40(3):495-510.

4. Fan XM, Huang CL, Wang YM, Li N, Liang QL, Luo GA. Therapeutic Effects of Tangshen Formula on Diabetic Nephropathy in db/db Mice Using Cytokine Antibody Array. J Diabetes Res. 2018: 8237590.

5. Do MH, Hur J, Choi J, Kim Y, Park HY, Ha SK. Spatholobus suberectus Ameliorates Diabetes-Induced Renal Damage by Suppressing Advanced Glycation End Products in db/db Mice. 2018. Int J Mol Sci. 19(9): 2774. 

6 Lai X, Tong D, Ai X, et al. Amelioration of diabetic nephropathy in db/db mice treated with tibetan medicine formula Siwei Jianghuang Decoction Powder extract. Sci Rep. 2018;8(1):16707.

7. Wu JS, Shi R, Lu X, Ma YM, Cheng NN. Combination of active components of Xiexin decoction ameliorates renal fibrosis through the inhibition of NF-κB and TGF-β1/Smad pathways in db/db diabetic mice. PLoS One. 2015;10(3):e0122661.

8. Sharma BR, Kim HJ, Rhyu DY. Caulerpa lentillifera extract ameliorates insulin resistance and regulates glucose metabolism in C57BL/KsJ-db/db mice via PI3K/AKT signaling pathway in myocytes. J Transl Med. 2015;13: 62.

9. Kim OK, Nam DE, Jun W, Lee J. Cudrania tricuspidata water extract improved obesity-induced hepatic insulin resistance in db/db mice by suppressing ER stress and inflammation. Food Nutr Res. 2015;59:29165.

Reviewer 2 Report

This manuscript described Dianthus superbus improves glomerular fibrosis and renal dysfunction in diabetic nephropathy model. I could not comments on results of HPLC. What is the explanation for Fig 8A shows DS-EA are only effective at 10ug/ml, while Fig 8B shows they are effective at 5ug/ml. and Why 10ug/ml treatment shows a higher Collagen IV than 5ug/ml treatment.

Author Response

I am very much thankful to the reviewers for their deep and thorough review. I hope my revision has improved the paper to a level of their satisfaction. I revised the text in the Methods, Results, Discussion, and reference (red colored part).

 Reviewer 2

This manuscript described Dianthus superbus improves glomerular fibrosis and renal dysfunction in diabetic nephropathy model. I could not comments on results of HPLC. What is the explanation for Fig 8A shows DS-EA are only effective at 10ug/ml, while Fig 8B shows they are effective at 5ug/ml. and Why 10ug/ml treatment shows a higher Collagen IV than 5ug/ml treatment.

 - As a result of further experiments, Collagen IV mRNA expression was effective at DS-EA 10ug/ml concentrations. Therefore, we corrected the Fig 8B data.

Round  2

Reviewer 1 Report

The authors have addressed all my comments except for the comment 3. i.e., Control mice treated with the low-dose and high-dose of the extract (db/m+DSL and db/m+DSH) is missing in the experimental design. These two groups are crucial for comparing the effects of DSL and DSH in db/db mice. 

As stated earlier, a concurrent control to tease out the effects of low-dose and high-dose of the extract on control mice is crucial to substantiate the effects of DS extract against diabetic nephropathy. 

Based on findings shown in Figure 4, it is clear that the DS extract possesses anti-hyperglycemic properties and improves insulin sensitivity, which would reduce the progression of DN. Nevertheless, the findings from studies in mesangial cells (shown in Figure 9) indicates that DS extract has direct anti-fibrotic and anti-inflammatory effects against Ang II-mediated induced the renal mesangial cell proliferation and significantly enhanced TGF-β/Smad signaling and collagen IV levels. Hence, I recommend including a few sentences detailing the direct and indirect effects (through improvements in glycemic profile) of DS extract against DN in the conclusion.

Author Response

 I am very much thankful to the reviewers for their deep and thorough review. I hope my revision has improved the paper to a level of their satisfaction. We revised the text in the Conclusion (red colored part) and revised the manuscript according to the iThenticate report.

 Reviewer 1

 The authors have addressed all my comments except for the comment 3. i.e., Control mice treated with the low-dose and high-dose of the extract (db/m+DSL and db/m+DSH) is missing in the experimental design. These two groups are crucial for comparing the effects of DSL and DSH in db/db mice.

 As stated earlier, a concurrent control to tease out the effects of low-dose and high-dose of the extract on control mice is crucial to substantiate the effects of DS extract against diabetic nephropathy.

 Based on findings shown in Figure 4, it is clear that the DS extract possesses anti-hyperglycemic properties and improves insulin sensitivity, which would reduce the progression of DN. Nevertheless, the findings from studies in mesangial cells (shown in Figure 9) indicates that DS extract has direct anti-fibrotic and anti-inflammatory effects against Ang II-mediated induced the renal mesangial cell proliferation and significantly enhanced TGF-β/Smad signaling and collagen IV levels. Hence, I recommend including a few sentences detailing the direct and indirect effects (through improvements in glycemic profile) of DS extract against DN in the conclusion.

 We have included the direct and indirect effects (through improvements in glycemic profile) of DS extract against DN in the conclusion part.